# Resistance Profile of Bovine Mastitis Isolates, Presence of the *mec*A Gene and Identification of *ESBL* Producing Strains from Small Rural Dairy Properties

**DOI:** 10.3390/ani13071147

**Published:** 2023-03-24

**Authors:** Kawany Gabrieli Zanetti Fazoli, Laisa Marina Rosa Rey, Kariny Aparecida Jardim Rúbio, Mateus Augusto Garcia Souza, Halison Murilo da Silva Oliveira, Débora Cristina Ribeiro, Kelly Regina de Jesus Duarte Pereira, Denise Miyuki Kawamo, Thays Karollyni Amaral Gomes, Isadora Brito da Silva, Isabela Carvalho dos Santos, Larissa Rafaela de Paula Ferreira, Isabelle Luiz Rahal, Juliana Silveira do Valle, Suelen Pereira Ruiz, Maria Graciela Iecher Faria, Zilda Cristiani Gazim, Ranunlfo Piau Junior, Daniela Dib Gonçalves

**Affiliations:** 1Graduate Program in Animal Science with Emphasis on Bioactive Products, Universidade Paranaense (UNIPAR), Umuarama 87502-210, PR, Brazil; 2School of Veterinary Medicine, Universidade Paranaense (UNIPAR), Umuarama 87502-210, PR, Brazil; 3Graduate Program in Medicinal and Phytotherapeutic Plants in Primary Care, Universidade Paranaense (UNIPAR), Umuarama 87502-210, PR, Brazil; 4Graduate Program in Biotechnology Applied to Agriculture, Universidade Paranaense (UNIPAR), Umuarama 87502-210, PR, Brazil

**Keywords:** antibiotic, bacterial multidrug resistance, prevention, one health

## Abstract

**Simple Summary:**

Bovine mastitis is an inflammation of the mammary gland in response to invasion by opportunistic agents. The objective of this work was to identify and evaluate the antibacterial resistance profile of mastitis milk samples, milking hands and milking equipment from small dairy farms belonging to northwest region of the state of Paraná, Brazil. Fifteen small, non-technical dairy farms in three municipalities, all belonging to the northwest region of the state of Paraná, Brazil, were selected. Of the 199 samples collected from the 15 selected properties in the municipalities of Boa Esperança, Tapejara and Juranda, 36.20% were classified as multiresistant. It is also worth noting the presence of an isolate of *Enterobater agglomerans* and one of *Moellerella wisconsensis* in the hands of milkers and milking machines, phenotypically producing extended-spectrum beta-lactamase (ESBL). As for the presence of the *mec*A gene, 72.72% isolated came from milk, 18.18% from insufflators and 9.1% from milking hands. Mastitis can be spread to the herd through the milking process by the milkers’ instruments and hands and adequate management measures can prevent its transmission and the conscious use of antibiotics decreases the prevalence of multidrug-resistant pathogens. The results of this work directly reflect on the health of the animals, the health of the workers and the health of the respective environment, which can make possible the continuity of the propagation of the etiological agents involved in the mastitis infection.

**Abstract:**

Bovine mastitis is an inflammation of the mammary gland in response to invasion by opportunistic agents. Due to the high economic importance of dairy production and the complexity related to animal health, the objective of this work was to identify and evaluate the antibacterial resistance profile of samples of mastitis milk, milking hand and milking equipment from small rural dairy farms belonging to the northwest region of the state of Paraná, Brazil. Five small, non-technical dairy farms in the municipalities of Boa Esperança, Juranda and Tapejara, all belonging to the northwest region of the state of Paraná, Brazil, were selected. The properties had Holstein and/or crossbred herds, carried out a bucket-by-foot milking system and all had the presence of animals with subclinical mastitis confirmed by the California Mastitis Test. Samples of sterile swabs from the milking insufflators, the milking hand and milk samples were collected—and later, isolation tests and phenotypic characterization of the samples, sensitivity tests to antimicrobials and phenotypic tests for the detection of beta-producing strains were performed with extended-spectrum beta-lactamase (ESBL), molecular identification of *Staphylococcus aureus* isolates and *mec*A gene research. Of the 199 samples collected from the 15 selected properties in the municipalities of Boa Esperança, Tapejara and Juranda, 72 (36.20%) were classified as multiresistant. Isolated from milkers’ hands and milking machines, which phenotypically produce extended-spectrum beta-lactamase (ESBL), the presence of the *mec*A gene was also observed in 11 isolates of *Staphylococcus* spp. of milk samples, machines and milking hands. Mastitis can be spread to the herd through the milking process by the milkers’ instruments and hands, and adequate management measures can prevent its transmission and the conscious use of antibiotics decreases the prevalence of multidrug-resistant pathogens. In this work, different pathogenic bacteria were detected in mastitic milk, milking equipment and milking hand with a high percentage (36.20%) of isolates classified as multidrug resistant. In addition, the presence phenotypically (ESBL) and molecularly (*mec*A gene) of isolates carrying resistance genes was also verified. These results directly reflect on the health of the animals, the health of the workers and the health of the respective environment, which can enable the continuity of the propagation of the etiological agents involved in the mastitis infection. The awareness of producers and workers on these properties about the disease, transmission, sanitary aspects and adequate management and treatment are essential for improving milk production and production efficiency.

## 1. Introduction

Brazilian cattle raising has great prominence in world dairy production, responsible for the production of 34 billion liters per year, generating high financial turnover and employment, with more than 415,000 people make a living from the activity in the country and about 170,000 considered small producers, who milk up to 250 L per day [1,2].

Bovine mastitis generates high economic losses. It is estimated that the economic losses caused by mastitis, from its clinical manifestation to its cure, range from $60.23 to $123.50 per cow [3]. As a result, there is a reduction in milk production, expenses with medicines, veterinary assistance, decrease in milk quality and disposal of contaminated milk and animals with chronic infection [4].

Bovine mastitis is an infectious disease characterized by inflammation of the mammary gland in response to the invasion of opportunistic agents, mainly caused by bacteria [5]. This disease can manifest itself in a subclinical, clinical or chronic form, which can severely compromise the general condition of the animal [6].

The clinical form causes visible changes in the udder, evident signs of inflammation, associated with changes in milk composition [7]. The subclinical is characterized by an asymptomatic infection or only a decrease in milk production can be observed [8]. In the chronic form, there is fibrosis—with an absence of signs of the inflammatory process and changes in the milk, the result of a persistent process—which may form fistulas in the affected mammary gland [9].

The microorganisms involved in the infection can be from contagious or environmental source [9]. The contagious ones live and multiply inside the mammary gland and their transmission occurs horizontally, during the milking process, through milking machines that indicate hygiene failure, through the hands of milkers and multiple-use towels [10]. On the other hand, environmental microorganisms live in the environment where the animals are raised and their transmission occurs between milkings, through contact of the teats with the contaminated environment, favoring the penetration of pathogens into the mammary gland [11].

The main factors that contribute to the spread of the disease in the herd are related to the history of the presence of mastitis and treatments used, type of milking, failure to evaluate the first three jets of milk, failures in teat disinfection and drying, incorrect cleaning and maintenance (and use) of equipment and lack of hygiene of the milkers’ hands [12].

Due to the high economic importance of dairy production and the complexity related to animal health, the aim of this study was to identify and evaluate the antibacterial resistance profile of mastitis milk samples, milkers’ hands and milking equipment from small rural dairy farms belonging to the northwest region of the state of Paraná, Brazil.

## 2. Material and Methods

### 2.1. Criterion for Inclusion of Rural Properties, Origin and Number of Samples

Five small, non-technical dairy farms were selected, with a bucket milking system, intended for dairy production with Holstein and/or Crossbred herds, with the presence of animals with subclinical mastitis confirmed by the CMT test (California Mastitis Test). Eight to ten animals were included in this research, with one milker and two samples of milking machines before and two after the start of milking on each rural property located in the municipalities of Boa Esperança, Juranda and Tapejara, all belonging to the northwest region of the state of Paraná, Brazil.

### 2.2. Sample Collection

From May to June 2022, samples were collected from the milking machines and the milker’s hand using sterile swabs containing AIMES medium + Activated Charcoal (Copan Transystem©, Brescia, Italy), which were collected through rotational movements of the swabs in the insufflators of the milking machine and the hands of the milkers in the respective places.

Before proceeding with the collection of milk from the animals, they were submitted to the CMT and the animals diagnosed with two crosses in the respective tests were selected to collect the milk samples using sterile flasks; the extraction was performed manually.

The milkers were also invited to complete the epidemiological questionnaires of the property and the milker for a possible explanation of the procedures performed in the milking routine, hygiene and maintenance of equipment, personal hygiene of the milker, along with the questionnaire, and monitoring of milking was carried out to observe the important points for the spread of mastitis in the herd.

After collection, the biological materials were kept under refrigeration for a period not exceeding 24 h and sent to the Laboratory of Preventive Veterinary Medicine and Public Health of the Graduate Program in Animal Science with Emphasis on Bioactive Products at Universidade Paranaense (UNIPAR) for processing and the carrying out of diagnostic techniques.

In total, 199 biological samples were collected—45, 42 and 35 milk samples, respectively—from the municipalities of Boa Esperança, Juranda and Tapejara. Five samples from the milker’s hand in each municipality and 20, 20 and 22 samples from milking machines, respectively, from the municipalities of Boa Esperança, Juranda and Tapejara were collected, totaling 122 milk samples, 15 hand samples from milkers and 60 samples from milking machines.

### 2.3. Culture, Isolation and Phenotypic Characterization of Samples

Samples were inserted into brain heart infusion (BHI) medium and incubated at 37 °C for 24 h. Then, they were sown using the depletion technique on plates containing blood agar medium and incubated at 37 °C for up to 48 h for bacterial isolation.

The predominant isolate was selected (where the bacterial colony was visually in greater quantity on the plate), and it was submitted to the analysis of the macroscopic and microscopic characteristics and catalase and coagulase tests, allowing the classification into positive coagulase *Staphylococcus* (CoPS) and coagulase negative *Staphylococcus* (CoNS) [13,14].

The biochemical identification of bacteria belonging to the Order Enterobacteriales was performed using the “Kit for Enterobacteria” (NewProv^®^, Pinhais, PR, Brazil), according to the manufacturer’s recommendations.

### 2.4. Antimicrobial Sensitivity Tests

Antimicrobial susceptibility tests were performed according to the criteria of the Clinical and Laboratory Standards Institute. The antibiotics were chosen based on the European Medicines Agency’s “Categorization of antibiotics for use in animals for prudent and responsible use”. Bacterial isolates using the disk diffusion test were evaluated against Amoxicillin + Clavulanate (30 µg), Amikacin (30 µg), Amoxicillin (10 µg), Cefoxitin (30 µg), Clindamycin (2 µg), Chloramphenicol (30 µg), Ceftiofur (30 µg), Doxycycline (30 µg), Erythromycin (15 µg), Meropenem (10 µg), Oxacillin (1 µg) and Rifampicin (5 µg). All strains classified as intermediate were considered non-susceptible as determined by the World Health Organization. Strains that showed resistance to three or more classes of antimicrobials were considered multidrug resistant (MDR) [15].

### 2.5. Phenotypic Test for Detection of Extended Spectrum Beta-Lactamase (ESBL) Producing Strains

The phenotypic test for the detection of enterobacteria producing extended-spectrum beta-lactamases (ESBL) was performed by the synergic double disc test with cefotaxime (30 μg), ceftazidime (30 μg), ceftriaxone (30 μg) and aztreonam (30 μg). The disks were distributed at a distance of 20 mm from a disk containing amoxicillin + clavulanate (20/10 μg). Any increase or distortion of the inhibition zone of one of the antibiotics toward the amoxicillin + clavulanate disk was considered suggestive of ESBL production [16].

### 2.6. Molecular Identification of Staphylococcus aureus Isolates

Polymerase chain reaction (PCR) was performed on coagulase positive *Staphylococcus* (CoPS) samples to verify which of these isolates were *Staphylococcus aureus*. The DNA was extracted with the PurelinkGenomic DNA Kit (Invitrogen, Carlsbad, CA, USA) according to the manufacturer’s information and the reactions were performed using the primer Sa442-1 (5′-AAT CTT TGT CGG TAC ACGATA TTC TTC ACG-3′ and the primer Sa442-2 (5′-CGT AAT GAGATT TCA GTA GAT AAT ACA ACA-3′) following the methodology [17]. For DNA amplification, an AppliedBiosystems Veriti™ 96-Well ThermalCycler (Waltham, MA, USA) was used.

The amplification of the products was visualized by electrophoresis on a 2% agarose gel stained with Gel Red (Uniscience, Osasco, SP, Brazil) using a molecular marker of 100 pb and the products were visualized as a single band of 241 pb.

### 2.7. mecA Gene Research

The DNA of *Staphylococcus* spp. classified as resistant to oxacillin was extracted using the Purelink Genomic DNA Kit (Invitrogen, Carlsbad, CA, USA) according to the manufacturer’s information and the PCR reactions were performed using the mecA1 primer (AAAATCGATGGTAAAGGTTGG) and mecA2 (AGTTCTGCAGTACCGGATTTG) [18]. For DNA amplification, an AppliedBiosystems Veriti™ 96-Well Thermal Cycler (Waltham, MA, USA) was used.

The amplification of the products was visualized by electrophoresis in a 2% agarose gel stained with Gel Red (Uniscience, Osasco, SP, Brazil) using a molecular marker of 100 pb and the products were visualized as a single band of 533 pb.

## 3. Results

A total of 199 samples were collected from the 15 selected properties, 70 from the municipality of Boa Esperança, 67 from the municipality of Juranda and 62 from Tapejara.

Regarding the bacterial growth of the samples, the municipality of Boa Esperança had the highest percentage among the three, with 71.43% (50) of the samples with growth (Table 1), followed by the municipality of Tapejara with 56.45% (35) samples (Table 2), and Juranda with 47.76% (32) of the samples with bacterial growth (Table 3).

Regarding microorganisms, in the municipality of Boa Esperança, non-*aureus* CoPS (12.85%), *Staphylococcus aureus* (22.85%) and *Escherichia coli* (5.71%) were detected. In the municipality of Tapejara, non-aureus CoPS (6.45%), *Staphylococcus aureus* (4.53%) and *Escherichia coli* (6.45%) were also detected. In the municipality of Tapejara, non-*aureus* CoPS (6.45%), *Staphylococcus aureus* (4.53%) and *Escherichia coli* (6.45%) were also detected. In the municipality of Tapejara, non-*aureus* CoPS (6.45%), *Staphylococcus aureus* (4.53%) and *Escherichia coli* (6.45%) were also detected. In the municipality of Juranda, non-*aureus* CoPS (5.97%), *Staphylococcus aureus* (2.98%) and *Escherichia coli* (19.40%) were detected.

In these municipalities, there was also the presence of isolates resistant to the antibiotics tested mainly against oxacillin, with resistance of 78%, 64.86% and 81.25%, respectively, in the municipalities of Boa Esperança, Tapejara and Juranda, which also showed resistance to clindamycin in 44%, 72.97% and 62.5%, respectively, in addition to other antibiotics with significant numbers of resistant isolates, such as erythromycin, cefoxitin and rifampicin (Table 4).

Among the 199 (100%) isolates, 72 (36.20%) were classified as multidrug resistant—that is, resistant to at least 1 drug from 3 or more classes of antimicrobials [17]. Of this total, 43 (59.72%) came from equipment used in milking, 20 (27.78%) came from milk and 9 (4.52%) came from the hands of milkers.

It is also worth mentioning, within these results, the presence of an isolate of *Enterobacter agglomerans* and one of *Moellerella wisconsensis*, having been, respectively, isolated from the hand of a milker and an insufflator, phenotypically producing ESBL (Figure 1).

In addition to the presence of phenotypically resistant isolates, the presence of the *mec*A gene was found in 11 of the *Staphylococcus* spp. isolates, 8 (72.72%) from milk, 2 (18.18%) from milking machine and 1 (9.1%) from the hand of a milker (Table 5 and Table 6).

## 4. Discussion

Mastitis is an endemic and economically important pathology, mainly in family farms, which are farms with low production; milkers do not have technical knowledge and guidance from trained professionals.

Related to bacterial growth, the municipality of Boa Esperança had the highest percentage, among the three municipalities studied, with 71.43% of the samples with growth. Some authors, such as [5], described the main points for mastitis control as the correct hygiene and periodic maintenance of milking equipment. Ideally, it should be checked every six months of its operation, in addition to changing the hoses that come into contact with the milk and the milking machines, as well as the hygiene of the milker and the use of gloves.

The results demonstrate the lack of hygiene and the failure to carry out periodic maintenance of equipment used in the milking process and possibly have contributed to the spread of mastitis-causing agents in the herd, because the insufflators of the teat cup sets were dirty before and after milking, as well as their maintenance only being carried out when there is a defect in their operation. Furthermore, most properties go years without carrying out this maintenance, in addition to the presence of poor hygiene (presence of mud and feces) in the facilities where the animals remained after milking.

In addition to the situations above, this high bacterial growth may also be related to the failure to pre-dip and dry the animals’ teats. The teats were dirty even after carrying out the pre-dipping and the drying of the teats was not carried out completely. In turn, failure in this management can lead to contamination of the insufflators of the teat cup sets, and consequently, contribute to the dissemination of different pathogenic agents, as already described by [10].

Another factor that may have contributed to the increase in this bacterial growth was the lack of hygiene of the milkers (due to lack of knowledge). It was observed that when starting the milking process they did not have the habit of washing their hands, and even during the process, they maintained contact with different body parts of the animals and handled the ropes and gates before returning to the milking routine, a situation that could favor cross-contamination (man × animal × environment). A study, conducted by Shin et al. [11], points out that the personal hygiene of the milker associated with adequate handling, especially during milking, can decrease the number of animals affected by clinical and subclinical mastitis, reduce the rate of new infections, improve the somatic cell count (SCC) of the herd and the quality of the milk produced.

Algharib et al. [1] point out that recommending the training of milkers on good practices for performing milking, principles of personal hygiene and the correct use of milking equipment are essential for controlling mastitis in the herd. Another factor that can be considered relevant to the result of this study is that most of the milkers in that municipality were over 50 years old and had been exercising this activity for many years; however, they did not seek and evolve in terms of obtaining correct instructions/technical information to improve their work routine, milk production and, consequently, the health of the herd.

The municipality of Tapejara was the second municipality with the highest bacterial growth in the collected samples, with 56.45% of the samples, followed by Juranda with 47.76%. This decrease in bacterial growth can be highlighted by better hygiene of the equipment used in the milking process. From the beginning and end of the milking process, the equipment was clean, in addition to the observed efficiency in carrying out pre-dipping and drying from the ceilings. When starting the milking process, milkers wash their hands and avoid touching dirty utensils. In these two municipalities, most properties received private veterinary technical assistance and producers have a basic technical understanding of good practices in milk production.

In this study, a common situation in all properties of the three municipalities is the non-adoption of the milking line, which possibly may have favored the spread of mastitis to the herd. This must be instituted to avoid the transmission of pathogenic agents from an infected animal to healthy animals—that is, healthy animals must be milked first, followed by animals with mastitis, a situation that is also reported in studies by [5].

In the municipality of Boa Esperança and Tapejara, Non-*aureus* CoPS, 12.85% and 6.45%, respectively, and *Staphylococcus aureus*, 22.85% and 4.53%, were isolated in the municipality of Juranda for Non-*aureus* CoPS 5.97% and *Staphylococcus aureus* 2.98% was isolated less frequently. These agents are associated with subclinical mastitis of contagious origin—that is, they are easily disseminated by the herd, they are passed unnoticed because they do not present evident clinical signs and in some cases a decrease in milk production [8,11].

The Non-*aureus* CoPS treatment is carried out in drying and has good responses to conventional dry cow therapies, since *Staphylococcus aureus* is the causative agent of mastitis with the lowest percentage of cure, due to its protective barrier, which is the ability to form an abscess inside the mammary gland, making the action of antibiotics difficult, in addition to being methicillin resistant bacteria, their treatment is carried out in the drying process with the use of an intramammary antibiotic associated with an injectable [6,12].

Non-aureus CoPS and *Staphylococcus aureus* can be controlled or even stopped from herd transmission through the adoption of the milking line, starting milking with animals negative for mastitis in the dark bottom cup test and in the CMT, followed by animals with mastitis or by the method of disinfecting the insufflators with disinfectants. When milking the animal before moving on to the next one, the insufflators can be sprayed or immersed in the disinfectant [3,4]. Another very important method for the control and eradication of this agent is good personal hygiene practices, such as washing hands before starting the milking process and whenever touching utensils and dirty objects or the use of gloves are measures that reduce the spread of this agent by the herd [1].

Authors [19] isolated 436 samples of raw milk from animals with clinical and subclinical mastitis from 3 farms and identified 135 samples of *Staphylococcus aureus*. Additionally, Ref. [20] isolated 23 samples of *Staphylococcus aureus* from a total of 48 samples of bovine milk diagnosed with subclinical mastitis—which are similar results to this work, which featured isolated *Staphylococcus aureus* from (16/70) samples from the municipality of Boa Esperança, (3/62) samples from Tapejara and (2/67) samples from Juranda.

Authors [21] evaluated in their work 1,549 milk samples from 952 cows, including cows with recurrent mastitis. Non-*aureus Staphylococcus* (NaS) (27.6%) was isolated, followed by *Escherichia coli* (18.9%) and *Staphylococcus aureus* (7.7%) of milk samples. In this study, it was possible to isolate 12.85% of samples with non-*aureus* CoPS from the municipalities of Boa Esperança, 6.45% from Tapejara and 5.97% from Juranda, followed by 22.85%, 4.53% and 2.98%, respectively, positive for *Staphylococcus aureus* (7.7%).

Researchers [22] collected 400 samples of bovine subclinical mastitis milk and in their analysis of 173 (43.25%) *E*. *coli* isolates was detected, which corroborates this work where in the municipality of Juranda it was the one that presented the highest percentage of *Escherichia coli*, with (13/67) 19.40%, followed by Tapejara with (3/62) 6.45% and Boa Esperança with (4/70) 5.71%.

*Escherichia coli* is one of the causative agents of environmental clinical mastitis, they are bacteria that live in the environment, so the main way to prevent mastitis caused by this agent is to keep the place where the animals are destined after milking clean, free of manure and sludge [5,19].

Mastitis caused by *Escherichia coli* has clinical signs such as a drop in milk production, fever, dehydration, severe depression, loss of appetite and signs of inflammation in the udder, which can lead to severe infections and even death of the animal [7]. Mastitis caused by *Escherichia coli* is unlikely to cure spontaneously, requiring treatment with antibiotics and cephalosporins which have been widely used in the treatment of mastitis, including cephalexin and ceftiofur. Other widely used antibiotics are amoxicillin, erythromycin, gentamicin and penicillin. Animals that are vaccinated against strains of *Escherichia coli* have a better cure rate for treatments with antibiotics [4,8].

Related to antimicrobial susceptibility, in the properties of the three municipalities of this study, bacterial isolates resistant to different antibiotics were detected, mainly against oxacillin (with resistance of 78%, 64.86% and 81.25%), clindamycin (44%, 72.97% and 62.5%) and erythromycin (9.95%, 15.67% and 11.33%) in the municipalities of Boa Esperança, Tapejara and Juranda, respectively. Furthermore, Dea et al. [3] also showed in their study antimicrobial susceptibility in 202 isolates of *S*. *aureus* from samples of mastitic bovine milk and only identified resistance to penicillin (12.4%) and erythromycin (0.5%); however, they did not detect resistance to oxacillin, a situation that differs from the results of this work.

It was also identified, in this work, a higher percentage of resistance to clindamycin in 44%, 72.97% and 62.5%, respectively, in the municipality of Boa Esperança, Tapejara and Juranda. Another important antibiotic with important numbers of resistant isolates was erythromycin.

Molineri et al. [12] aimed to determine the prevalence of phenotypic resistance of *S*. *aureus* to antimicrobial agents collected worldwide, in the context of bovine intramammary infections between the years 1969 and 2020, whereby the highest global prevalence of resistant *S*. *aureus* was for the penicillin class, followed by clindamycin, erythromycin and gentamicin. The ceftiofur and cephalothin class showed the lowest global prevalence of antimicrobial resistance, highlighting the results obtained in our work.

Different classes of antibiotics are used to treat animals with mastitis. The most commonly used first-line treatments are penicillins (oxacillin, amoxicillin, methicillin), alone or associated with aminoglycosides (amikacin, gentamicin, streptomycin, neomycin), macrolides (erythromycin, azithromycin), lincosamides (clindamycin), fluoroquinolones (norfloxacin), tetracyclines (doxycycline, oxytetracycline) and cephalosporins (ceftiofur) [4,10,11].

Antimicrobial resistance is the ability of a microorganism to grow or survive in the presence of an antimicrobial at a concentration that is generally sufficient to inhibit or kill microorganisms of the same species [20]. Nader et al. [6] reported in their work that the indiscriminate use of antimicrobials, with the wrong concentration, dosage or application interval, has increased and accelerated this resistance process, making it a serious threat to public health. One of the main causes of antimicrobial resistance in dairy cattle is the fact that it is associated with the use of antibiotics for the treatment of mastitis, whether in the lactation phase or in the drying process of the animal, associated with its erroneous use in the production [12].

Dea et al. [3] justify that the increase in antimicrobial resistance in mastitis-causing agents is caused due to the attempt to treat mastitis conditions without knowing which agent caused the infection, thus the treatment in the dark develops new agents with genetic mutation making them more resistant the drugs of choice used. The conscientious use of antibiotics in the production line, such as the use of the active ingredient of antibiotics that is effective and recommended for the causative agent, in addition to dosage, concentration and correct period, are methods of prevention that mitigate the emergence of new causative agents of genetically resistant mastitis [7].

Molineri et al. [12] also analyzed all antimicrobials which showed a growing pattern over time, being more evident from 2009 that the antimicrobials with the highest prevalence of resistance over the years were clindamycin, gentamicin and oxacillin, which is similar to the results of this study where oxacillin and clindamycin were the most prevalent.

In this study, a result that draws attention is the prevalence of 72 (36.20%) bacterial isolates classified as multi-drug resistant. Of this total, 45 (62.50%) come from the equipment used in milking, 18 (9.04%) from the milk and 9 (4.52%) from milk workers’ hands.

In our study, 18 (9.04%) multiresistant samples isolated from milk were identified and a similar result was detected [20], where they isolated 48 *Staphylococcus aureus* from samples of mastitis bovine milk and all of them were multi-drug resistant. Related to the presence of resistant multidrug isolates [4], they also detected it in different species of *Streptococcus*.

Taniguchi et al. [21] isolated *Listeria* spp. of 79 (26.3%) of the 300 samples, including 29 (36.7%), 32 (40.5%) and 18 (22.8%) isolates found in raw milk, milking equipment and hand swabs of milkers, respectively, with 88% of the total multidrug resistant isolates, a result similar to our work, where 72 (36.20%) samples were classified as multidrug resistant from milk sample isolates, swabs from milking equipment and milkers’ hands.

It is also worth mentioning the presence of an isolate of *Enterobacter agglomerans* and one of *Moellerella wisconsensis*, having been isolated from milking hand and insufflator hand, respectively, phenotypically producing ESBL [21], isolated from 1549 milk samples from mastitis cows and 952 of the samples with bacterial growth. The incidence of ESBL-producing extended-spectrum β-lactamase (ESBL) was 1.4% of all samples and 1.4% of *Klebsiella pneumoniae* enterobacteria were identified. Additionally [23] evidenced in their study of mastitis milk samples the occurrence of *Escherichia coli* (118/372, 31.7%) and *Klebsiella pneumoniae* (77/372, 20.7%), two environmental pathogens known to cause bovine mastitis. When searching for extended-spectrum beta-lactamases (ESBLs) agents were detected in selective medium in (3/118, 1.59%) *Escherichia coli* and (6/77, 7.79%) *Klebsiella pneumoniae*, emphasizing the results obtained in our work. Enterobacteriaceae are more frequently present in properties with precarious hygienic-sanitary conditions, with a lot of fecal contamination, accumulation of manure and sludge [22].

The inappropriate use of antimicrobials can lead to the emergence of resistant strains. Enterobacteria are among the main etiological agents of environmental bovine mastitis and are often resistant to antimicrobials, especially to β-lactams due to the production of beta-lactamases and some enterobacteria produce broad-spectrum beta-lactamases (ESBL), constituting the bacteria the ability to degrade β-lactams, in addition to presenting broad spectrum on various antimicrobials such as ceftazidime, cefotaxime and aztreonam, which are strongly inhibited by clavulanic acid [4,10].

Haenni et al. [10] suggest that cows should have access to feed immediately after milking, in order to keep them standing until the teat dries and the striated canal closes completely, preventing possible contamination after milking by agents from the environment. This method is carried out in all three municipalities—the animals are kept standing if fed for more than an hour after milking.

*mec*A gene expression is constituted or induced by beta-lactam antibiotics, such as oxacillin and cefoxitin. The *mec*A gene is inserted into the staphylococcal chromosome through a mobile genetic element called the chromosomal staphylococcal cassette. The *mec*A gene sequence is highly conserved in strains of *Staphylococcus aureus* and *Staphylococcus* spp. negative coagulase. The *Staphylococcus* spp. are pathogens that cause bovine mastitis and may present multiple resistance to different antimicrobial groups.

Carvalho et al. [24] aimed in their study to phenotypically identify isolates of *Staphylococcus* spp. obtained from bovine milk and to characterize its antimicrobial resistance profile. Of the 101 strains isolated, the *mec*A gene was detected in 27% of the milk samples. In our studies, similar results were obtained, which noted the presence of the *mec*A gene in 11 of the isolates of *Staphylococcus* spp., 8 (72.72%) isolated from milk, 2 (18.18%) from insufflators and 1 (9.1%) from the hand of a milker. Other studies also detected the presence of the *mec*A gene in milk, which showed the importance of this gene for human, animal and environmental health [25,26,27].

A study conducted by Dea et al. [3] recommended the treatment of dry cows, with the aim of reducing subclinical infections and preventing new infections in the dry period that occurs between two lactations, which is the phase in which the animal was in productive rest, an important situation for the health of the animal’s mammary gland to regenerate for the next lactation. This can last an average of 60 days, counting from drying to delivery. In the first post-drying weeks, the risk rate for new infections is very high and the treatment of subclinical mastitis has higher cure rates compared to treatment during lactation. According to the study carried out in this work, all properties in the three municipalities adopt drying therapy, which helps in the control and treatment of possible subclinical infections in animals.

One of the main mastitis control methods is dry cow therapy, adopting the milking line, keeping cows standing after milking, washing the milk pool correctly and the application of a treatment protocol, effectiveness in the pre- and post-dipping procedure and the use of gloves by the milkers. An important role of veterinarians is to advise on the importance of mastitis in the production line, the economic losses it entails, how to prevent it and the correct treatment. Furthermore, much of this assistance is oriented around management planning to maintain or improve the health status of the family properties.

## 5. Conclusions

In this study, different pathogenic bacteria were detected in mastitis milk, milking equipment and milker’s hand with a high percentage (36.20%) of isolates classified as multidrug resistant. In addition, the presence phenotypically (ESBL) and molecularly (*mec*A gene) of isolates carrying resistance genes were also verified. These results directly reflect on the health of the animals, the health of the workers and the health of the respective environment, which can make possible the continuity of the propagation of the etiological agents involved in the mastitis infection. The awareness of producers and workers on these properties about the disease, transmission, sanitary aspects and adequate management and treatment are essential for improving milk production and production efficiency.

## Figures and Tables

**Figure 1 animals-13-01147-f001:**
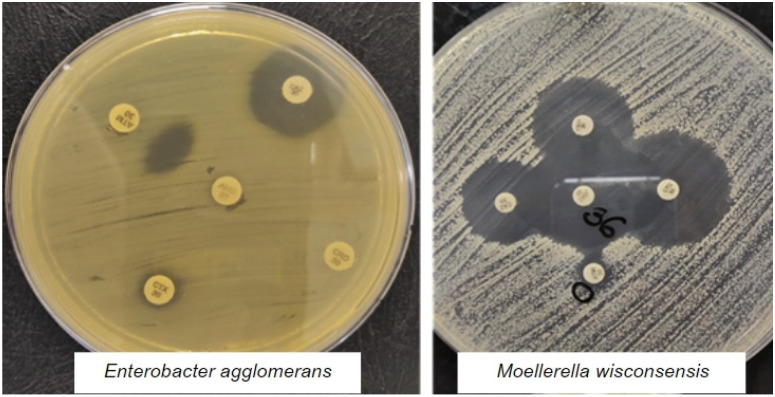
Phenotypic test for the detection of positive ESBL-producing strains, showing an increase or distortion in the zone of inhibition of one of the antibiotics (Cefotaxime, Ceftazidime, Ceftriaxone and Aztreonam) toward the amoxicillin + clavulanate disc of an isolate of *Enterobacter agglomerans* and one of *Moellerella wisconsensis* isolated from the hand of a milker and an insufflator, respectively.

**Table 1 animals-13-01147-t001:** Frequency of microorganisms isolated in milk samples, hand swabs from milkers and milking machines, in the municipality of Boa Esperança, northwest of the State of Paraná, Brazil, May–June 2022.

Microorganisms	Milk	Milkers’ Hand	Milking Machines	N°	%
CoNS	12	-	2	14	20
CoPS Non-*aureus*	1	1	7	9	12.85
*Staphylococcus aureus*	9	3	4	16	22.85
*Escherichia coli*	3	-	1	4	5.71
*Enterobacter aerogenes*	-	-	1	1	1.44
*Enterobacter agglomerans*	1	1	1	3	4.28
*Kluyvera ascorbata*	2	-	-	2	2.85
*Moellerella wisconsensis*	-	-	1	1	1.44
No growing	17	-	3	20	28.58
Total	45	5	20	70	100

Subtitle: CoNS: Coagulase-negative *Staphylococcus*; CoPS: Coagulase-Positive *Staphylococcus*-Non-*aureus*.

**Table 2 animals-13-01147-t002:** Frequency of microorganisms isolated in milk samples, hand swabs of milkers and milking machines, in the municipality of Tapejara, northwest of Paraná State, Brazil, May–June 2022.

Microorganisms	Milk	Milkers’ Hand	Milking Machines	N°	%
CoNS	9	3	8	20	31
CoPS-Non-*aureus*	4	-	-	4	6.45
*Staphylococcus aureus*	3	-	-	3	4.53
*Escherichia coli*	-	-	4	4	6.45
*Citrobacter koseri*	-	-	1	1	1.60
*Kluyvera ascorbata*	-	1	1	2	3.22
*Moellerella wisconsensis*	-	-	1	1	1.60
*Serratia marcescens*	-	1	-	1	1.60
No growing	19	-	7	26	43.55
Total	35	5	22	62	100

Subtitle: CoNS: Coagulase-negative *Staphylococcus*; CoPS: Coagulase-Positive *Staphylococcus*-Non-*aureus.*

**Table 3 animals-13-01147-t003:** Frequency of microorganisms isolated from milk samples, hand swabs from milkers and milking machines, in the municipality of Juranda, northwestern Paraná State, Brazil, May–June 2022.

Microorganisms	Milk	Milkers’ Hand	Milking Machines	N°	%
CoNS	10	1	-	11	16.41
CoPS-Non-*aureus*	4	-	-	4	5.97
*Staphylococcus aureus*	1	-	1	2	2.98
*Escherichia coli*	-	3	10	13	19.40
*Enterobacter aerogenes*	-	-	1	1	1.50
*Hafnia alvei*	-	-	1	1	1.50
No growing	27	1	7	35	52.24
Total	42	5	20	67	100

Subtitle: CoNS: Coagulase-negative *Staphylococcus*; CoPS: Coagulase-Positive *Staphylococcus*-Non-*aureus.*

**Table 4 animals-13-01147-t004:** Resistance profile of enterobacteria isolated from milk samples, hand swabs from milkers and milking machines, from the municipality of Boa Esperança, Tapejara and Juranda, northwest of Paraná State, Brazil, May–June 2022.

Bacterial Resistance					
Antibiotics	Milk	Milkers’ Hand	Milking Machines	N°	%
Amoxicillin + clavulanate	0	1	5	6	3.10
Amikacin	0	0	6	6	3.10
Amoxicillin	2	3	13	18	9.33
Cefoxitin	4	3	14	21	10.89
Clindamycin	4	6	21	31	16.06
Chloramphenicol	0	2	10	12	6.21
Ceftiofur	1	0	7	8	4.15
Doxycycline	1	0	2	3	1.55
Erythromycin	2	4	17	23	11.92
Meropenem	0	0	2	2	1.04
Oxacillin	6	5	22	33	17.10
Rifampicin	5	5	20	30	15.55
TOTAL	25	29	139	193	100

**Table 5 animals-13-01147-t005:** Resistance profile of *Staphylococcus* spp. *mec*A gene carriers, from the municipality of Boa Esperança, northwestern Paraná State, Brazil, May–June 2022.

Samples		Antibiotics and Resistance												Presence of the *mec*A Gene
		AMC	AMI	AMO	CFO	CLI	CLO	CTF	DOX	ERI	MER	OXA	RIF	
Animals	Identification													
1	*S. aureus*	S	S	S	S	S	S	S	S	S	S	R	R	P
3	CoNS	S	S	S	S	S	S	S	S	S	S	S	S	P
13	CoNS	S	S	S	S	S	S	S	S	S	S	R	S	P
14	CoNS	S	S	S	S	S	S	S	S	S	S	S	S	P
Milking Machines														
After 4/2	*S. aureus*	S	R	S	R	I	R	I	I	S	S	R	I	P
After 5/1	*S. aureus*	S	S	I	S	R	S	S	S	R	S	R	R	P

Subtitle: I—Intermediate, R—Resistant, S—Sensitive, P—Positive, N—Negative, After—Swabs collected after the milking process, Before—Swabs collected before starting the milking process, AMC- Amoxicillin + Clavulanate, AMI—Amikacin, AMO—Amoxicillin, CFO—Cefoxitin, CLI—Clindamycin, CLO—Cloxacillin, CTF—Ceftiofur, DOX—Doxycycline, ERI—Erythromycin, MER—Meropenem, OXA—Oxacillin, RIF—Rifampin. CoNS: Coagulase-negative *Staphylococcus*.

**Table 6 animals-13-01147-t006:** Resistance profile of *Staphylococcus* spp. *mec*A gene carriers, from the municipality of Tapejara, northwestern Paraná State, Brazil, May–June 2022.

Samples		Antibiotics and Resistance												Presence of the *mec*A Gene
		AMC	AMI	AMO	CFO	CLI	CLO	CTF	DOX	ERI	MER	OXA	RIF	
Animals	Identification													
70	CONS	S	S	S	S	R	S	S	S	R	S	R	S	P
77	CONS	S	S	S	R	S	S	S	S	S	S	S	S	P
78	CONS	I	S	R	R	R	S	S	S	R	S	R	R	P
80	CONS	S	S	S	S	R	S	S	S	R	S	R	R	P
Hands														
9	CONS	I	R	I	R	R	S	R	S	S	S	R	R	P

Subtitle: I—Intermediate, R—Resistant, S—Sensitive, P—Positive, N—Negative, AMC- Amoxicillin + Clavulanate, AMI—Amikacin, AMO—Amoxicillin, CFO—Cefoxitin, CLI—Clindamycin, CLO—Cloxacillin, CTF—Ceftiofur, DOX—Doxycycline, ERI—Erythromycin, MER—Meropenem, OXA—Oxacillin, RIF—Rifampicin, CoNS—Coagulase-negative *Staphylococcus*.

## Data Availability

The data presented in this study are available on request from the corresponding author. The data are not publicly available due to ethical and privacy issues.

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
