# Peer review of "Resistance Profile of Bovine Mastitis Isolates, Presence of the mecA Gene and Identification of ESBL Producing Strains from Small Rural Dairy Properties"

_animals, 2023, doi:10.3390/ani13071147_

Round 1

Reviewer 1 Report

The manuscript entitled „Resistance Profile of Bovine Mastitis Isolates, Presence of the 2 mecA Gene and Identification of ESBL Producing Strains From 3
Small Rural Dairy Properties
” reflects useful and innovative research that has a lot of potential going forward.

Overall, the manuscript represents cutting-edge research, fairly well organized, and carried out. Both the science and the presentation are strong. So, I would suggest that it be accepted for publication after minor corrections.

Reference

In the line 122 CLSI - Clinical and Laboratory Standards Institute (2019) should be in brackets [], and missing in part of REFERENCE

In the line 125, (EMA, 2019) should be in brackets [], and missing in part of REFERENCE

In the line, 129 (WHO, 2019) should be in brackets [], and missing in part of REFERENCE

In the line 257 A study by [12], should be replaced with Study conducted by Shin et al. [12]

In the line 262 [1] should be replaced with Algharib et al. [1].

In the line 338 Other study [3] should be replaced with Moreover Dea et al. [3]

In the line 346 [13], aimed should be replaced with Molineri et al. [13] aimed

In the line 360 [6], reported should be replaced with Nader et al. [6] reported

In the line 367 Research [3], should be replaced with Dea et al. [3]

In the line 375 [13], also analysed should be replaced with Molineri et al. [13] also analysed

In the line 388 Researches [23], should be replaced with Taniguchi et al. [23]

In the line 413 Research [11], should be replaced with Haenni et al. [11]

In the line 430 A study by [3] should be replaced with A study conducted by Dea et al. [3]

Correct abbreviated names of Journal

In line 467 correct abbreviated names of Journal Drug Deliv into Drug Deliv.

The majority of abbreviated names of Journal are not correct abbreviated (Please cheek any abbreviated names of Journal and correct according Journal rules)

In lines 505 and 507 name of Journal isn’t abbreviated.  Name of JournalJournal of Clinical Microbiology” should be abbreviated as J. Clin. Microbiol.

In line 511 name of Journal “Journal Clinical Microbiology and Infection” should be abbreviated as J. Clin. Microbiol. Infect.

Author Response

Dear Reviewer,

All considerations requested by the evaluators were accepted and corrected (in red) in the text. The manuscript underwent an English review by a specialized company (review statement attached).

The following were corrected: the authors' full names, the simple abstract was inserted, the abstract was reorganized, three more references were inserted and corrections were also made throughout the text.

Thank you for the opportunity to review this article in this esteemed magazine.

Any other need I am available

Thanks

Reviewer 2 Report

Dear colleagues,

I have read the manuscript entitled ‘Resistance Profile of Bovine Mastitis Isolates, Presence of the 2 mecA Gene and Identification of ESBL Producing Strains From 3 Small Rural Dairy Properties’ thoroughly.

The manuscript is discussing an interesting issue including both Gram-positive and Gram-negative pathogens from mastitis cases with associated antibacterial resistance profiles.

Additionally, the possibility of spread of mastitis infection in herds through hands of milkers and the adequate management procedures that have to be taken in consideration.

Overall, the manuscript is well written. The methods are thorough and well presented. Results and discussion are relevant. I don’t have a huge amount of comments and I agree to publish after minor language check.

Best regards

Author Response

(The authors gave the same response as above.)

Reviewer 3 Report

All corrections are in the revised manuscript

Author Response

(The authors gave the same response as above.)

Round 2

Reviewer 3 Report

The authors did not respond to most comments in the previous revision